# The association of denture wearing with reduced lung function and increased airflow limitation in 58–72 year old men

Niamh Kelly[1], Kyra Gormley[1], Dermot A. Linden[2], Lewis Winning[3], Mary McClory[2], Fionnuala T. Lundy[2], Kathy M. Cullen[4], Gerard J. Linden[5], Ikhlas A. El Karim[2]*

1 Centre for Dentistry, School of Medicine Dentistry and Biomedical Sciences, Queen's University Belfast, Belfast, United Kingdom, 2 Wellcome-Wolfson Institute for Experimental Medicine, School of Medicine, Dentistry and Biomedical Sciences, Queen's University Belfast, Belfast, United Kingdom, 3 Division of Restorative Dentistry and Periodontology, Dublin Dental University Hospital, Trinity College Dublin, Dublin, Ireland, 4 Centre for Medical Education, School of Medicine Dentistry and Biomedical Sciences, Queen's University Belfast, Belfast, United Kingdom, 5 Centre for Public Health, School of Medicine, Dentistry and Biomedical Sciences, Queen's University Belfast, Belfast, United Kingdom

* i.elkarim@qub.ac.uk

## Abstract

### Objective

To investigate the association between denture wearing and airflow limitation in men in Northern Ireland enrolled in the Prospective Epidemiological Study of Myocardial Infarction (PRIME) study.

### Methods

A case-control design was used to study partially dentate men. Cases were men aged 58–72 years who were confirmed as denture wearers. Controls were never denture wearers who were matched by age (± 1 month) and smoking habit to the cases. The men had a periodontal assessment and completed a questionnaire detailing their medical history, dental history and behaviours, social circumstances, demographic background and tobacco use. Physical examination and spirometry measurements of forced expiratory volume in one second ($FEV_1$) and forced vital capacity (FVC) were also undertaken. Spirometry data for edentulous men who wore complete dentures were compared with that recorded for the partially dentate men studied.

### Results

There were 353 cases who were partially dentate and were confirmed denture wearers. They were matched for age and smoking habit to never denture wearer controls. The cases had an $FEV_1$ that was on average 140 ml lower than the controls, $p = 0.0013$ and a 4% reduction in percent predicted $FEV_1$, $p = 0.0022$. Application of the GOLD criteria indicated that 61 (17.3%) of the cases had moderate to severe airflow limitation compared with 33 (9.3%) of controls, $p = 0.0051$. Fully adjusted multivariable analysis showed that partially dentate men who were denture wearers were significantly more likely ($p = 0.01$) to have

**Data Availability Statement:** All relevant data are within the manuscript and its Supporting information files.

**Funding:** this work was funded by the Northern Ireland Research and Development fund as grants RRG 1.2 and RRG 5.22 (PI: Linden). The funding body has since been renamed as the HSC Research and Development Division, Public Health Agency, Northern Ireland (Web address: research. hscni.net). The funders had no role in study design, data collection and analysis, decision to publish, or preparation of the manuscript.

**Competing interests:** The authors have declared that no competing interests exist.

moderate to severe airflow reduction with an adjusted odds ratio (OR) of 2.37 (95% confidence intervals 1.23–4.55). In the 153 edentulous men studied moderate to severe airflow limitation was recorded in 44 (28.4%), which was significantly higher than in the partially dentate denture wearers ($p = 0.017$), and the men who had never worn a denture ($p<0.0001$).

## Conclusion

Denture wearing was associated with an increased risk of moderate to severe airflow limitation in the cohort of middle-aged Western European men studied.

## Introduction

Associations between oral health and various respiratory diseases have been recognised for some time [1, 2]. Strong evidence and biological plausibility exists for an association with nosocomial hospital-acquired pneumonia [3, 4]. A recent systematic review concluded that pathogenic bacterial species identified in the mouth are associated with a higher incidence of aspiration pneumonia in older people in residential care [5].

Poor oral health has also been implicated in chronic obstructive pulmonary disease (COPD) [3, 6]. COPD is characterised by persistent reduction and accelerated rate of decline in forced expiratory volume ($FEV_1$). The disease is associated with an enhanced chronic airway inflammatory response to noxious particles or gases [7]. It has a worldwide prevalence of ~10% in adults >40 years of age and is now the third most common cause of global mortality [8] such that in 2015 COPD was responsible for 3.2 million deaths worldwide, an increase of 12% compared with 1990 [9].

A systematic review of fourteen observational studies reported a strong association between poor oral health and COPD that was attenuated but remained significant after adjustment for smoking [6]. The association between poor oral health and COPD is mostly based on studies that investigated periodontal disease as an exposure. A recent systematic review and meta-analysis supported an association between periodontitis and COPD [10]. Population-based studies have reported associations between periodontitis and reductions in measurements of lung function [11–13].

A potential unifying mechanism for a linkage between poor oral health and lung disease could relate to the close relationship between the respective microbiomes particularly as the oral microbiota can contribute to the lung microbiome [14–16]. The oral microbiome is influenced by factors in the oral microecosystem including saliva, various anatomic structures and the hard surfaces presented by the teeth [14]. In this context dentures, used to replace some or all of the teeth, form an important addition to the oral microecosystem and will influence the composition of the oral microbiome [17]. Denture plaque biofilms contain both bacterial species identified as opportunist respiratory pathogens [18] and *Candida albicans*, a yeast that has been shown to facilitate the growth of such pathogens [19]. Denture wearing could, theoretically at least, contribute to dysbiotic changes in both the oral and lung micro- and myco-biomes.

The Adult Dental Health Survey of the United Kingdom found that about one fifth of the population were denture wearers, including 13% of the population who relied on a combination of dentures and natural teeth [20]. We examined the hypothesis that denture wearing could be associated with lung function decline. The aim was to investigate the link between denture wearing and airflow limitation in a homogenous group of 58–72 year-old West European men. The study included both partially dentate and edentulous men who wore dentures.

## Materials and methods

### Dentate men

The study that focused on partially dentate men was a case-control design, nested within a cohort study. The men studied were participants in the Prospective Epidemiological Study of Myocardial Infarction (PRIME), a cohort study of cardiovascular disease in Northern Ireland. The sampling frame was based on industry, the civil service, and general medical practices. Between 1991 and 1994 a sample of 2745 representing approximately 5% of 50–60 year-old men from the greater Belfast area were recruited to match broadly the social class structure of the population in Northern Ireland [21].

From 2001 to 2003, surviving men were invited to attend for re-screening. At this point in the PRIME study a dental component was added to investigate possible associations between oral and dental health and incident cardiovascular and other systemic diseases. The case-control study is based on data collected from 1400 dentate men who had a comprehensive dental examination, completed a dental questionnaire and had spirometry during their re-screening visit. The re-screening took place in a research clinic in the School of Dentistry, Queen's University, Belfast. Approval for the project was obtained from the Research Ethics Committee of the Faculty of Medicine, Queen's University, Belfast, and the Office for Research Ethics Committees (Northern Ireland). The aims of the investigation and the nature of the study were fully explained to the subjects, who gave their informed written consent before participation.

### Cases and controls

The presence of a denture or dentures was confirmed at the time of the dental examination. Each partially dentate man who was a denture wearer was asked when he had first been fitted with a denture. This may have been a single upper or lower partial denture, or both. Alternatively it may also have been an upper or lower complete denture, as long as there were still natural teeth in the opposing arch. The cases represented all the men who met these criteria for denture wearing.

Men who did not have a denture at examination were asked the following question, taken from the 1988 Adult Dental Health Survey of the United Kingdom [22]: 'Have you ever had a denture fitted'? A negative response was taken to confirm that they had never worn a denture. The controls were never denture wearers who were matched by age (± 1 month) and smoking habit to the cases.

### Edentulous men

There were 158 men who had no teeth at the re-screening visit and all these men reported that they wore complete upper and lower dentures. Valid spirometry data were available for 155 of these men.

### Spirometric assessment

Spirometry was performed, at the re-screening visit, using a wedge bellows spirometer (Model S Vitalograph, Buckinghamshire, UK). The research nurses involved in spirometry measurement were trained by Respiratory staff at the Royal Victoria Hospital, Belfast. All measurements were performed in accordance with the American Thoracic Society / European Respiratory Society criteria [23]. To standardise the assessment men with dentures were instructed to wear their dentures during spirometry. The % predicted $FEV_1$ was obtained by applying the widely used equation of the European Community for Steel and Coal [24].

Airflow limitation was identified using GOLD criteria [25] in men who had $FEV_1/FVC <0.7$. The affected men were further subdivided into a mild group with % predicted $FEV_1 \leq 80\%$ and a moderate to severe group ($<80\%$). None of the men investigated were in the very severe category ($FEV_1 <30\%$).

## Definition of variables

At the re-screening visit each participant completed a questionnaire that gathered information on their demographic and socioeconomic background, level of education, tobacco consumption, and dental behaviour. Men aged over 64 years were categorised as old. Third molars were excluded from the periodontal assessment and were not included when the number of teeth was recorded. A high number of teeth equated to >20. The periodontal examinations were completed by one of four dental hygienists who had been calibrated against a "gold standard" set by a senior clinical researcher (GL) prior to the study. Regular monthly meetings took place to ensure inter- and intra-examiner consistency and reproducibility. Throughout the study, the hygienists maintained the standard set at the outset with κ values of >0.8 at the regular training sessions [26]. Clinical attachment level (CAL) was recorded as the distance from the cement–enamel junction (CEJ) to the base of the clinical pocket. This was calculated by measuring the distance from the CEJ to the gingival margin and subtracting this value from the probing depth measurement (recession was recorded as a negative value) [26, 27]. High CAL equated to a mean value >2mm. Plaque was measured at the time of dental examination using the Silness and Löe index [28] and high plaque equated with a score of >0.7. Self-reported dental attendance pattern was recorded as 'regular' or 'irregular'. Toothbrushing frequency was categorized as high equating to two or more times per day or low at less than twice per day. Socio-economic conditions were categorised into high, middle and low based on three proxy indicators: the type of living accommodation (rented or owned/mortgage), number of cars/vans/motorcycles in the household and the number of baths and/or showers and toilets in the home [29]. Education was assessed by the number of years in full-time education and a high level equated with >10 years. Body weight (to the nearest 200g) and height (to the nearest cm), were measured by research nurses trained and calibrated according to the PRIME protocol. BMI was calculated as weight/height$^2$ (kg/m$^2$). The BMI measured between 2001 and 2003 was categorised using the World Health Organization [30] classification: normal weight equated to BMI $<25$ kg/m$^2$, overweight $\leq 25$ kg/m$^2$ to $\geq 30$ kg/m$^2$ and obese $>30$ kg/m$^2$. Participants who reported that they had smoked more than 100 cigarettes were questioned about their smoking history. Smoking was categorised by self-report as current, past or never. Diabetes was categorised by self-report of the condition.

## Statistical analysis

Comparisons of baseline characteristics were made using the independent samples t-test for continuous variables and the Chi-square test for categorical variables. Multivariable analysis was carried out using multiple logistic regression to investigate the association between denture wearing and airflow limitation as the dependent variable in the partially dentate men. The OR was adjusted for BMI; diabetes; dental attendance; toothbrushing frequency; plaque level; education level; socioeconomic status; CAL, number of teeth, age and smoking. Data from the edentulous were compared with the partially dentate denture wearers and the never denture wearers. The level of statistical significance was set at $p < 0.05$. Analyses were performed using SPSS version 27 (IBM Corp., Armonk, NY, USA).

## Results

### Partially dentate denture wearers

In the case-control study the cases were 353 dentate men who were confirmed denture wearers and they were matched for age and smoking habit to never denture wearers who acted as controls. The cases had fewer teeth ($p<0.0001$), higher mean CAL ($p<0.0001$) and higher mean plaque scores ($p<0.0001$) than the controls (Table 1). The cases were also less likely to be regular dental attenders ($p = 0.02$) or to have a high education level ($p = 0.007$).

The cases had an $FEV_1$ that was on average 140 ml lower than the controls, $p = 0.0013$ and a 4% reduction in percent predicted $FEV_1$, $p = 0.0022$ (Table 2). Application of the GOLD criteria indicated that 17.3% (n = 61) of the cases had moderate to severe airflow limitation compared with 9.3% (n = 33) of controls, $p = 0.0051$ (Table 2).

Multivariable analysis showed that men who wore dentures were significantly more likely ($p = 0.0017$) to have moderate to severe airflow reduction with a crude unadjusted OR of 2.07 (95% CI 1.31–3.26). In the fully adjusted model the OR increased slightly to 2.37 (95% CI 1.23–4.55) and remained significant ($p = 0.01$). The only other variable that was significant in the fully adjusted model was smoking habit with both past (OR = 1.94, 1.05–3.57, $p = 0.035$) and current smoking (OR = 4.72, 2.44–9.13, $p<0.0001$) being associated with moderate to severe airflow reduction.

**Table 1. Oral, dental and non-dental factors by denture wearing status.**

|  |  | Cases | Controls | *p* |
|---|---|---|---|---|
|  |  | (n = 353) | (n = 353) |  |
| Age, years, mean (SD) |  | 64.64 (3.0) | 64.56 (2.9) | 0.74 |
| Teeth, mean (SD) |  | 13.20 (5.6) | 22.69 (3.6) | <0.0001 |
| Clinical attachment loss (mm), mean (SD) |  | 3.09 (1.5) | 2.05 (0.9) | <0.0001 |
| Plaque index, mean (SD) |  | 1.01 (0.5) | 0.69 (0.5) | <0.0001 |
| Dental attendance, n (%) |  |  |  |  |
|  | Regular | 222 (62.9%) | 251 (71.1%) | 0.02 |
| Toothbrushing, n (%) |  |  |  |  |
|  | High frequency | 190 (54.8%) | 215 (61.3%) | 0.08 |
| Smoking, n (%) |  |  |  |  |
|  | Never | 125 (35.4%) | 125 (35.4%) | * |
|  | Former | 155 (43.9%) | 155 (43.9%) |  |
|  | Current | 73 (20.7%) | 73 (20.7%) |  |
| BMI, n (%) |  |  |  |  |
|  | Normal | 89 (25.2%) | 82 (23.2%) | 0.06 |
|  | Overweight | 175 (49.6%) | 204 (57.8%) |  |
|  | Obese | 89 (25.2%) | 67 (19.0%) |  |
| Diabetes, n (%) |  |  |  |  |
|  | Yes | 25 (7.1%) | 15 (4.2%) | 0.10 |
| Education level, n (%) |  |  |  |  |
|  | High | 149 (42.2%) | 185 (52.4%) | 0.007 |
| Socio-economic conditions, n (%) |  |  |  |  |
|  | Low | 121 (34.4%) | 103 (29.2%) | 0.11 |
|  | Middle | 93 (26.4%) | 84 (23.8%) |  |
|  | HIgh | 138 (39.2%) | 166 (47.0%) |  |

Significant differences highlighted in bold.

* matched for smoking.

**Table 2. Lung function by denture wearing status.**

|  | Cases | Controls |  |
|---|---|---|---|
|  | (n = 353) | (n = 353) | *p* |
| $FEV_1$, L, mean (SD) | 2.68 (0.6) | 2.82 (0.6) | **0.0013** |
| Percent predicted $FEV_1$, Mean (SD) | 85.90 (17.83) | 89.88 (16.48) | **0.0022** |
| FVC, L, mean (SD) | 3.61(0.7) | 3.71 (0.7) | 0.09 |
| Airflow limitation, n (%) |  |  |  |
| Moderate to severe | 61 (17.3%) | 33 (9.3%) | **0.0051** |
| Mild | 21 (5.9%) | 17 (4.8%) |  |
| None | 271 (76.8%) | 303 (85.8%) |  |

Significant differences highlighted in bold.

## Edentulous denture wearers

The $FEV_1$ of edentulous men, who wore complete upper and lower dentures, was on average 2.46 (SD 0.6) L, which was significantly lower (*p* = 0.0002) than the partially dentate denture wearers. The percent predicted $FEV_1$ was 80.55 (SD 18.0), which was also significantly lower than both partially dentate denture wearers (p = 0.002) and men who had never worn a denture (*p*<0.0001). Moderate or severe airflow limitation was recorded in 28.4% (n = 44) and mild in 5.2% (n = 8) of the edentulous denture wearers and this was significantly higher than in the partially dentate denture wearers (*p* = 0.017) and the men who had never worn a denture (*p*<0.0001). Edentulous compared with partially dentate denture wearers had an OR of 1.90 (95%CI 1.21–2.97), *p* = 0.005 for moderate to severe airflow limitation, after adjustment for age and smoking. The corresponding value for edentulous versus dentate men who had never worn a denture was OR = 3.92 (95%CI 2.37–6.49), *p*<0.0001.

The edentulous men reported that they had first been provided with a denture at 37.4 (SD 12.3) years which was significantly younger (*p* = 0.001) than the partially dentate denture wearers at 44.0 (SD 15.5) years. The edentulous had been denture wearers for 28.3 (SD 12.2) years, which was significantly longer (*p*<0.0001) than the partially dentate denture wearers at 20.6 (SD 15.0) years.

## Discussion

The main finding of the population-based nested case-control study was that denture wearing, in men who retained some of their natural teeth, was associated with reduced lung function (as measured by $FEV_1$) and increased airflow limitation (as measured by a reduction in $FEV_1$/FVC ratio and application of the GOLD criteria). This relationship remained significant after adjustment for various known confounders. Further, the level of moderate to severe airflow limitation in edentulous men, who wore complete dentures, was even higher than partially dentate denture wearers. To the best of our knowledge this is the first report of a link between denture wearing and reduced lung function in a population-based study. Airflow limitation, which characterises chronic obstructive airway disease, is a major public health burden [31]. The identification of modifiable risk factors for obstructive lung disease could provide opportunities to modify the course of the disease and reduce its effects.

The edentulous men in the PRIME study, who wore complete dentures, had the poorest lung function with one in three having evidence of at least mild airflow limitation. By comparison slightly less than one in four of the partially dentate denture wearers had the same level of airflow limitation. This stepped relationship would further support an association with poorer

pulmonary function and may reflect the finding that the edentulous had been wearing dentures for almost 8 years longer, on average, than the partially dentate denture wearers. On a more positive note over recent years there has been a steady decline in the proportion of the population who have no natural teeth. Nevertheless, this will not immediately translate into a reduction in denture wearing [32]. A recent survey of adults attending dental practices in the UK reported that 13.7% had partial dentures; with a higher prevalence with increasing age so that more than 40% of those aged 75 or over were partial denture wearers [33]. Our understanding of the long-term consequences of complete and partial denture use is limited, particularly in relation any possible associations with systemic diseases or conditions [32, 34]. If, as suggested by the results of the present study, denture wearing in individuals who retain at least some of their natural teeth, could increase the risk of poorer lung function, then it will be important to assess the implications at a public health level.

The dental component of the Belfast PRIME study was primarily designed to investigate possible associations between periodontal and systemic diseases [27]. Summary data were collected on denture wearing and this allowed the secondary analysis of the PRIME database to test our hypothesis that there was an association between denture wearing and reduced lung function. The research hygienists ascertained those men who were currently wearing a denture at the re-screening visit and identified when they had first been provided with a denture. However, in the context of this epidemiological study the complexity of denture-use behaviour was not explored further and data were not collected on denture-related habits such as frequency of wear nor of denture hygiene procedures or denture cleanliness.

The association between denture wear and airflow limitation reported in this study does not provide information on possible mechanisms. Dentures support the growth of complex polymicrobial biofilms of bacteria and yeasts known as denture plaque [32, 35]. The majority (65%) of denture plaque biofilms are colonised by known respiratory pathogens, including *S. aureus*, *S. pneumonia*, *P. aeruginosa*, and *H. influenza*, leading O'Donnell and colleagues [18] to conclude that dentures can act as a reservoir for potential respiratory pathogens. The presence of such pathogens in denture plaque could be an important factor as the oral microbiota is a major source of the lung microbiome [14]. There has been recent interest in the association between the airway mycobiome and frequent exacerbations and mortality in COPD [36]. *Candida* spp. are secondary colonisers of denture plaque [32] and the most commonly identified species *Candida albicans* can facilitate the growth of respiratory pathogens [19] and increase the frequency of COPD exacerbations [37]. Further investigations of possible mechanistic links between alterations in both the oral and lung microbiomes, and mycobiomes, related to denture wearing and possible involvement in the pathogenesis of lung diseases would be helpful.

Beyond microbial dysbiosis, it is possible that dentures could influence pro-inflammatory signalling through exposure to toxic particles [38]. Acrylic resin materials, used to make dentures, are considered to be biologically acceptable but there is evidence that unpolymerised components and by-products of the polymerization reaction can produce harmful effects [39]. Denture components that leach and diffuse into the saliva can be swallowed or aspirated and could induce local as well as systemic effects [40, 41]. Whether this mechanism contributes to airflow limitation is currently not known and merits further investigation.

Complex interrelationships exist between the various oral and dental variables studied. Partially dentate denture wearers had higher plaque levels and a poorer periodontal condition than the controls. Irregular dental attendance, which was also evident in the cases, could have been a factor in the decision to choose extraction and replacement of teeth with a denture rather than to have treatment to retain teeth affected by caries or periodontal disease. The provision of a denture could therefore be a surrogate for poor dental health behaviours. Nevertheless, none of these factors were statistically significant in the final statistical model. The cases

and controls were matched for age and smoking habit as these represent two major risk factors for decline in lung function. It has been pointed out that matching in such a design does not eliminate confounding by the matching factors [42]. Accordingly, we adjusted for both age and smoking habit in our multivariate analysis. The final statistical model showed that current smoking was a strong independent predictor with a five-fold increased risk of moderate to severe airflow reduction. Past smoking was also significant in the fully adjusted model. These findings support smoking as a major environmental factor for reduced lung function.

The strengths of the PRIME study as outlined by Yarnell (1998) [21] include its size and the fact that it was completed on a representative sample of men in Northern Ireland. The participants were a well-characterised homogenous cohort of community-dwelling West European men with a limited age range. Lung function was measured by well-trained research nurses using standardised spirometry. The periodontal examination and the dental questionnaires were completed by experienced calibrated dental hygienists. Although the possibility of residual confounding cannot be excluded, the major covariates that could confound possible associations were accounted for in the regression analysis.

The study had a number of limitations. Firstly, it was conducted in a male only cohort and we therefore cannot draw conclusions on any potential associations in women. The PRIME study was designed in the late 1980s to investigate risk factors for the development of cardiovascular diseases, which were significantly more prevalent in men at that time [43]. Moreover, the limited age range of participants, same geographic location, and same ethnicity, additionally limit generalisability of findings. The study used a one-off spirometry measurement to assess lung function and it is therefore difficult to categorise the men as having chronic airflow limitation. It was not possible to identify whether the airflow limitation was due to COPD or asthma. The design of the study precluded the use of a bronchodilator and therefore a diagnosis of COPD could not be established. There was also limited information regarding how frequently the participants wore their denture(s), whether they slept with the denture in situ and there were no data collected on the denture cleaning regimens used. There was also limited information regarding how frequently the participants wore their denture(s), whether they slept with the denture in situ and there were no data collected on the denture cleaning regimens used. Although the comparison was primarily between denture wearers and non-wearers it is possible that denture quality, denture satisfaction and other factors related to removable prostheses could have influenced the results. It is acknowledged that the absence of such data was a limitation.

## Conclusions

Within the limitations of the study, the finding that denture wearing was associated with reduced $FEV_1$ and an increased risk of moderate to severe airflow limitation, in community-dwelling men, is novel. The findings highlight the need for investigations aimed at clarifying the possible impact of long-term denture wearing on lung function. This is particularly important given that despite a steady reduction in edentulousness nevertheless denture wearing is common and increases in frequency with increasing age. Strategies to improve oral hygiene, and in particular denture hygiene where appropriate, should be investigated as potential interventions that could help to reduce the burden of diseases associated with airflow limitation.

## Supporting information

**S1 Dataset. Minimum dataset.**
(XLS)

## Author Contributions

**Conceptualization:** Dermot A. Linden, Lewis Winning, Fionnuala T. Lundy, Gerard J. Linden, Ikhlas A. El Karim.

**Data curation:** Niamh Kelly, Kyra Gormley, Mary McClory, Kathy M. Cullen, Gerard J. Linden, Ikhlas A. El Karim.

**Formal analysis:** Lewis Winning, Gerard J. Linden.

**Funding acquisition:** Gerard J. Linden.

**Investigation:** Kathy M. Cullen.

**Methodology:** Lewis Winning.

**Supervision:** Ikhlas A. El Karim.

**Validation:** Ikhlas A. El Karim.

**Writing – original draft:** Fionnuala T. Lundy, Gerard J. Linden, Ikhlas A. El Karim.

**Writing – review & editing:** Niamh Kelly, Dermot A. Linden, Lewis Winning, Kathy M. Cullen.

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
