## [Decision Letter · Decision Letter 0]

22 Feb 2023

PONE-D-22-32347The association of denture wearing with reduced lung function and increased airflow limitation in 58-72 year old menPLOS ONE

Dear Dr. Elkarim,

Thank you for submitting your manuscript to PLOS ONE. After careful consideration, we feel that it has merit but does not fully meet PLOS ONE’s publication criteria as it currently stands. Therefore, we invite you to submit a revised version of the manuscript that addresses the points raised during the review process.

Please address the specific concerns raised by the reviewers and I look forward to receiving the revised manuscript. ==============================

We look forward to receiving your revised manuscript.

Kind regards,

Mirza Rustum Baig

Academic Editor

PLOS ONE

Journal Requirements:

3. "In your Data Availability statement, you have not specified where the minimal data set underlying the results described in your manuscript can be found. PLOS defines a study's minimal data set as the underlying data used to reach the conclusions drawn in the manuscript and any additional data required to replicate the reported study findings in their entirety. All PLOS journals require that the minimal data set be made fully available. For more information about our data policy, please see http://journals.plos.org/plosone/s/data-availability.

We will update your Data Availability statement to reflect the information you provide in your cover letter."

Reviewers' comments:

Reviewer's Responses to Questions

**Comments to the Author**

1. Is the manuscript technically sound, and do the data support the conclusions?

Reviewer #1: Partly

Reviewer #2: Yes

Reviewer #3: Yes

2. Has the statistical analysis been performed appropriately and rigorously? 

Reviewer #1: I Don't Know

Reviewer #2: Yes

Reviewer #3: Yes

3. Have the authors made all data underlying the findings in their manuscript fully available?

Reviewer #1: Yes

Reviewer #2: Yes

Reviewer #3: No

4. Is the manuscript presented in an intelligible fashion and written in standard English?

Reviewer #1: Yes

Reviewer #2: Yes

Reviewer #3: Yes

5. Review Comments to the Author

Reviewer #1: This is a well-written, well-structured and cohesive account. The study draws on the major strength of the Belfast PRIME study after it was broadened in scope in 2001-2003. The case-control design is wholly appropriate, and the authors are to be commended on exploring an important, and in a novel way, the potential association between denture wearing and airflow limitation/COPD. Indeed, from the title of the paper, the reader would be led to assume that the work will, in an unambiguous way, allow valid conclusions to be drawn in this regard. Unfortunately, the limitations that the authors correctly mention in the Discussion (lines 352-355, 406-408 and 418-422) are serious detractors from the strength of the present study as it relates to denture wearing per se. Certainly, oral and dental factors, and specifically periodontal parameters, were meticulously recorded, as was the purpose of the Belfast PRIME study, but the omission of key denture-related factors (subjective satisfaction, technical quality and denture retention/stability, denture wearing habits, denture cleanliness, and so on), each of which have potentially critical implications for airflow limitation, have not been accounted for and this seems to be a weakness that cannot at the stage be corrected.

Generally, however, the quality of the documented work is of a high order. A clarification is needed regarding the re-screening during 2001-2003 at which time surviving men (presumably from the 1991-1994 sample of n=2745 50-60 year old men) were drawn; but it is not clear how the present denture study sample of n=353 could be aged 58-72 years if they were from a 50-60 year-old group of 10 years prior...?

Some minor edits are needed: line 126, refs should be [14, 15]?; Table 1, "Material conditions" should be socioeconomic conditions? -

Whereas the authors do acknowledge the limitations of not including denture factors, one still feels that this omission has material interactions with outcomes. It seems odd how such a relatively simple set of data was not included in the otherwise detailed oral data collection.

Reviewer #2: The research work is original and highly interesting. The study is well designed, control group is convincing with the same age group and smoking habit.

The data analysis and the statistical method were appropriate.

The results were presented clearly with 2 test groups: partially and fully edentulous groups.

The discussion was thorough and convening.

The conclusion was formulated carefully and in appropriate manner.

Reviewer #3: The authors have presented a timely, technically sound and relevant manuscript addressing issues relevant to population aging and its challenges. The statistical analysis seems to have been performed appropriately and rigorously but i'm not an expert in this area. I'm not clear with regards to the full availability of the data related to this study as I only see all the relevant summary statistics in the manuscript. Ive attached some very minor review suggestions and comments in the attachment (highlighted in yellow and sahred in the comment box)

6. PLOS authors have the option to publish the peer review history of their article (what does this mean?). If published, this will include your full peer review and any attached files.

Reviewer #1: No

Reviewer #2: No

Reviewer #3: No

---

## [Author Response · Author response to Decision Letter 0]

27 Feb 2023

Please see attached file" response to reviewers"

---

## [Editor Report · Decision Letter 1]

16 Apr 2023

PLOS ONE Re: [PONE-D-22-32347] - [EMID:bd2f71e71b90fce6] The association of denture wearing with reduced lung function and increased airflow limitation in 58-72 year old men

PONE-D-22-32347R1

Dear Dr. Elkarim,

We’re pleased to inform you that your manuscript has been judged scientifically suitable for publication and will be formally accepted for publication once it meets all outstanding technical requirements.

Kind regards,

Mirza Rustum Baig

Academic Editor

PLOS ONE

Additional Editor Comments (optional):

Please add the statement on the effect of denture quality, denture satisfaction and other removable prostheses related factors which might have possibly influenced the results, although the comparison was primarily between denture wearers and non-wearers. This is a major limitation which needs to be clearly highlighted in the dicussion section of the study.  
---

## [Editor Report · Acceptance letter]

10 May 2023

PONE-D-22-32347R1 

The association of denture wearing with reduced lung function and increased airflow limitation in 58-72 year old men 

Dear Dr. El Karim :

I'm pleased to inform you that your manuscript has been deemed suitable for publication in PLOS ONE. Congratulations! Your manuscript is now with our production department. 

Kind regards, 

on behalf of

Dr. Mirza Rustum Baig 

Academic Editor

PLOS ONE